# Acute Wheel-Running Increases Markers of Stress and Aversion-Related Signaling in the Basolateral Amygdala of Male Rats

**DOI:** 10.3390/jfmk8010006

**Published:** 2022-12-30

**Authors:** Kolter B. Grigsby, Nathan R. Kerr, Taylor J. Kelty, Xuansong Mao, Thomas E. Childs, Frank W. Booth

**Affiliations:** 1Department of Biomedical Sciences, University of Missouri, Columbia, MO 65211, USA; 2Department of Behavioral Neuroscience, Oregon Health & Science University, and VA Portland Health Care System, Portland, OR 97239, USA; 3Department of Nutrition and Exercise Physiology, University of Missouri, Columbia, MO 65211, USA; 4Department of Physiology, University of Missouri, Columbia, MO 65211, USA; 5Dalton Cardiovascular Center, University of Missouri, Columbia, MO 65211, USA

**Keywords:** physical activity, amygdala, dynorphin, HPA axis, exercise, stress

## Abstract

Physical activity (PA) is a non-invasive, cost-effective means of reducing chronic disease. Most US citizens fail to meet PA guidelines, and individuals experiencing chronic stress are less likely to be physically active. To better understand the barriers to maintaining active lifestyles, we sought to determine the extent to which short- versus long-term PA increases stress- and aversion-related markers in wild-type (WT) and low voluntary running (LVR) rats, a unique genetic model of low physical activity motivation. Here, we tested the effects of 1 and 4 weeks of voluntary wheel-running on physiological, behavioral, and molecular measures of stress and Hypothalamic Pituitary Adrenal (HPA)-axis responsiveness (corticosterone levels, adrenal wet weights, and fecal boli counts). We further determined measures of aversion-related signaling (kappa opioid receptor, dynorphin, and corticotropin releasing hormone mRNA expression) in the basolateral amygdala (BLA), a brain region well characterized for its role in anxiety and aversion. Compared to sedentary values, 1, but not 4 weeks of voluntary wheel-running increased adrenal wet weights and plasma corticosterone levels, suggesting that HPA responsiveness normalizes following long-term PA. BLA mRNA expression of prodynorphin (*Pdyn*) was significantly elevated in WT and LVR rats following 1 week of wheel-running compared to sedentary levels, suggesting that aversion-related signaling is elevated following short- but not long-term wheel-running. In all, it appears that the stress effects of acute PA may increase molecular markers associated with aversion in the BLA, and that LVR rats may be more sensitive to these effects, providing a potential neural mechanism for their low PA motivation.

## 1. Introduction

There are major physiological, psychological, and social consequences of stress, including immune suppression [1,2,3], increased adiposity [4,5], heart disease [6,7,8], depression [9], and greater healthcare costs [10]. This is compounded by the largely overlapping negative health consequences of physical inactivity, the fourth leading cause of preventable death worldwide [11]. Moreover, populations experiencing continuous stress are less likely to be physically active compared to less stressed populations [12]. Healthy physical activity (PA) is considered a viable behavioral strategy for coping with subjective stress [13,14,15,16]; however, these benefits go largely unrealized by the ~97% of adult Americans who fail to meet US activity guidelines [17]. 

Acute PA, similar to other behavioral coping strategies (overeating, substance and alcohol use, etc.), is considered physiologically stressful. Widely adopted models of PA in rodents, such as treadmill running and forced swimming, are known to increase hypothalamic-pituitary-adrenal axis (HPA) activity and other measures of stress [18,19]. Although less stressful than forced models, acute wheel activity is similarly known to increase tissue sensitivity to glucocorticoids and increase plasma corticosterone levels, both of which suggest heightened HPA activity [20]. However, the stress-related responses to acute PA appear to adapt over time. For example, plasma corticosterone levels are known to peak following 1 week of wheel-running in male rats, and to return to baseline by 4-weeks [21]. Consequently, this adaptation coincides with reinforcement of wheel-running in rodents [22,23,24,25]. Using a well-characterized behavioral measure of reward, Greenwood et al. determined that male Sprague-Dawley rats did not develop a conditioned place preference (CPP; a measure of reward) for wheel-running at 2 weeks but did develop a preference at 6 weeks. Similarly, rats selectively bred for low voluntary wheel-running (LVR) behavior developed a preference for long-term wheel-running after 4 weeks, despite running ~5-fold less and showing lower motivation for voluntary wheel-access than their founders (female Wistar rats) [26]. Although it appears the adaptations to PA as a physiological stressor may play a critical role in its reinforcement and maintenance, the underlying neurobiological mechanisms remain poorly understood. More importantly, an improved understanding of these mechanisms could help identify potential barriers to exercise adherence in humans. 

Acute HPA activation increases aversion-related signaling in the brain, namely through the effects of Corticotropin Releasing Factor (CRF) acting on the Kappa Opioid Receptor (KOR) system. When injected at the level of the basolateral amygdala (BLA)—a brain region well characterized for its importance in aversion—KOR antagonists have been shown to reduce anxiety-like behavior and fear responses in rats [27]. KOR antagonists have gained interest as potential pharmacotherapies in the treatment of stress-related behavioral outcomes, namely depression and anxiety [28,29,30]. Substantial literature supports the importance of physical activity in reducing clinical signs of anxiety, depression, and subjective stress (for review, see [31,32,33]); however, few studies have directly addressed importance of stress- and aversion-related responses in the development and maintenance of PA. Therefore, the present study seeks to determine the physiological and neurobiological consequences of acute and chronic PA to better understand potential neurobiological barriers to PA adherence. 

To evaluate the stress effects of short-term versus long-term wheel-running, our current work attempts to address two sets of questions to determine (1) if wild-type (WT) and low voluntary wheel-running (LVR) rats, a genetic risk model of low physical activity motivation, show increased markers of stress (plasma corticosterone levels and adrenal wet weights) after 1 week of wheel-running, and whether those markers return to baseline after 4 weeks of wheel-running and (2) if markers of aversion-related signaling [Kor, Dynorphin, and CRF mRNA expression and Opioid Receptor Kappa 1 (OPRK1) protein phosphorylation—a marker of KOR activity] are elevated in the BLA after 1 week of wheel-running, and whether these levels return to baseline by 4 weeks. In addition, this experiment tests whether the low physical activity of LVR rats results from either elevated baseline markers of stress (corticosterone and adrenal wet weights), and/or a heightened stress reactivity (novel environment-induced fecal boli count) and aversion-related molecular signaling (KOR, Dynorphin, and CRF mRNA expression and KOR phosphorylation) in response to acute wheel-running. 

## 2. Materials and Methods

### 2.1. Animals and Experimental Procedure

Experimental protocols (protocol number 9320) were approved by the University of Missouri Animal Care and Use Committee on 26 February 2020. In total, 12–15-week-old male WT and LVR rats were maintained on a 12:12-h light/dark cycle at 21–22 °C. Food (Formulab Diet 5008, Purina, Hartwell, GA, USA) and water were provided ad libitum. All efforts were made to minimize suffering and reduce the number of animals used in this study. Male WT Wistar rats were purchased from Charles River Laboratories Inc (Houston, TX, USA). Male LVR rats were from generations 20–22 and were derived from a foundry population developed by Booth et al. [34]. In brief, LVR rats were selected for the primary phenotype of running low nightly distance based on 6 days of voluntary wheel-running (from 28–35 days of age) [34]. WT (n = 5–6/group) and LVR (n = 6/group) rats were single housed in standard cages with voluntary running wheels (circumference 1.062 m; Tecniplast 215F0105, Tecniplast, Italy) for either 1 or 4 weeks, where nightly running distance was recorded daily. A third group of WT (n = 5) and LVR (n = 6) rats remained sedentary (no wheel-access) for this period. At the end of the wheel-running experiments, rats were euthanized by CO_2_ asphyxiation. After which, blood, adrenal glands, and brains were rapidly removed and 2 mm diameter BLA punches were immediately frozen in liquid nitrogen and kept at −80 °C until processing. For this experiment, we tested the effects of 1 versus 4 weeks of wheel-running on the periphery by analyzing adrenal wet weights, plasma corticosterone, and fecal boli counts in a novel environment (a marker of emotionality and stress [35,36,37]) in WT and LVR rats. Regarding brain outcomes, we also tested whether 1 week of wheel-running increased mRNA expression of prodynorphin (*Pdyn*), kappa-opioid-receptor (*Kor*), or corticotropin releasing factor (*Crf*) in the BLA and whether these markers of aversion-related signaling returned to baseline by 4 weeks of wheel access. All experiments were done in the same set of animals discussed above and rats with wheel access retained their wheel access up until time of sacrifice.

### 2.2. Plasma Corticosterone Levels

Approximately 2 mL of blood was taken via cardiac puncture from the above WT and LVR rats within one-hour of lights-on, which is a known trough in diurnal plasma corticosterone levels [38,39]. Blood was placed in 2.5 mL LH lithium heparin separator tubes (Greiner Bio-one, Kremsmunster, Austria) and centrifuged at 5000× *g* for 15-min. The plasma was then stored at −80 °C until processing. Plasma corticosterone was measured using an ELISA kit (Item K014-H5, Arbor Assays, Eisenhower PI, Ann Arbor, MI, USA), which has a sensitivity of 20.9 pg ·mL^−1^. Following the manufacturer’s recommended protocol, 50 µL of plasma per animal were assayed in duplicate. Corticosterone levels were determined using the equation obtained from the standard curve plot and expressed as pg · mL.

### 2.3. Novel Environment Fecal Boli Count

To determine potential behavioral differences in the acute stress response, fecal boli counts were taken from sedentary WT and LVR rats following 20-min in a novel environment. Colonic transit time and fecal boli number have been associated with increased HPA activity [40,41]. Moreover, intra-hypothalamic CRF antagonism has been shown to greatly decrease stress-induced fecal output [40]. For testing, WT and LVR rats were blindly brought to an experimenter in a dimly lit room and gently placed into a novel chamber for 20 min [24]. The testing chamber measured 40 cm L × 30 cm W × 30 cm H, contained walls with black, horizontal stripes (2.5 cm wide, 2.5 cm apart), and had a textured dark gray floor. Fecal boli counts were recorded and the chamber was cleaned with 70% ethanol before the next test. Rats were euthanized 2 days later to minimize potential stress effects.

### 2.4. BLA mRNA Isolation and qPCR mRNA Analysis

BLA mRNA, from the above WT and LVR rats, was isolated as previously described [42,43]. mRNA was quantified using a Nanodrop 1000 (Thermo Fisher, Waltham, MA, USA), and lack of mRNA degradation was confirmed using a 1% agarose gel. cDNA synthesis was performed according to a previously established method [43]. mRNA levels of *Crf*, *Pdyn*, *Oprk1*, and *18S* were assayed in duplicate using SYBR-GREEN Mastermix (Bio-Rad, Hercules, CA, USA) and gene-specific primers (Applied Biosystem, Foster City, CA, USA) (Table 1).

### 2.5. Western Blotting for BLA Phospho-OPRK1 (KOR) Protein Expression

BLA protein extraction methods were performed as previously described by our lab [44]. In brief, 15–20 mg of BLA tissue was homogenized on ice in RIPA buffer (50 mM Tris-HCl (pH 8.0), 150 mM NaCl, 1% NP40, 0.5% sodium deoxycholate, 1% SDS, 1X protease inhibitor cocktail) using a Tissuelyser at 25 Hz for 1 min X2. Tissue homogenates were centrifuged (12,000× *g*), and supernatant protein concentrations were obtained via a BCA assay (Pierce Biotechnology, Rockford, IL, USA). In total, 60 µg of protein was loaded onto 12% SDS-Page gels. Proteins were then transferred onto nitrocellulose membranes and all blots were incubated with Ponceau S (Sigma, St Louis, MO, USA) to verify equal loading among all lanes. The primary antibody (rabbit polyclonal) for phospho-OPRK1 (1:1000, Thermo-Fisher, Cat# PA5-40216) was diluted in Tris-buffered saline + Tween20 with 5% bovine serum albumin and applied overnight at 4 °C. Horseradish peroxidase-conjugated secondary antibody (1:1000; CST, Cat# 7074), was applied for 1 hr at room temperature, and ECL substrate (Pierce Biotechnology) was then applied for 5 min prior to exposure. Band densitometry was determined using a Kodak 4000R Imager and Molecular Imagery Software (Kodak Molecular Imaging Systems, New Haven, CT, USA).

### 2.6. Statistical Analysis

Analyses were performed and graphs were prepared using GraphPad Prism version 8.2.1 (Graphpad Software, San Diego, CA, USA). All values are presented as mean ± standard error (SE); all data analyzed using ANOVA assumed a Gaussian distribution and passed the D’Agostino–Pearson normality test. Significance was set at an alpha value of 0.05. Main effects of strain (WT vs. LVR) and time (Sedentary, 1-week and 4-week conditions) on adrenal wet weights, corticosterone levels, and BLA mRNA expression (*Crf*, *Pdyn*, and *Kor*) were analyzed using a two-way Analysis of Variance (ANOVA). Sidak post-hoc comparisons were used to validate statistically significant interactions between factors for adrenal wet weights, plasma corticosterone levels, and *Crf*, *Pdyn*, and *Kor* mRNA expression. Novel environment fecal boli counts were determined using a two-tailed, Student’s t-test. pOPRK1 protein expression was normalized to ponceau total protein and analyzed using a one-way ANOVA followed by Sidak post-hoc analysis.

## 3. Results

### 3.1. Male WT Rats Run Greater Daily Distances Than Lvr Rats at 4, but Not at 1 Week of Wheel-Running

Average daily running distances at 1 and 4 weeks of wheel running for male WT (n = 5/cage condition) and LVR (n = 5–6/cage condition) rats are shown in Figure 1A–C. A two-way ANOVA for strain versus time revealed a significant interaction (F(1,17) = 6.074; *p* < 0.05), a main effect of strain (F(1,17) = 9.42; *p* < 0.01), and a main effect of time (F(1,17) = 13.12; *p* < 0.01). Unexpectedly, a Sidak post-hoc comparison showed no difference between WT and LVR running distance at 1 week (0.49 ± 0.13 km vs. 0.12 ± 0.03 km; *p* = 0.65), but a significant difference at 4 weeks (2.25 ± 0.65 km vs. 0.35 ± 0.04 km; *p* < 0.01). Because of the ~6-fold difference in running distance, 1- and 4-week running distances are shown separately for WT (Figure 1B) and LVR (Figure 1C) rats. Both WT and LVR rats had higher daily running distances at 4 weeks compared to 1 week of wheel-running, whereas WT rats ran ~4.5-fold further (0.49 ± 0.13 km vs. 2.25 ± 0.65 km; *p* < 0.01; Figure 1B) and LVR rats ran ~3-fold further (0.12 ± 0.03 km vs. 0.35 ± 0.04 km; *p* < 0.05; Figure 1C).

### 3.2. Male LVR Rats Display Higher Levels of Hpa Activity across Basal and Wheel Access Conditions

We next determined whether baseline differences in HPA activity markers existed between male WT and LVR rats under sedentary and/or wheel conditions. Fecal boli counts in a novel environment are a simple and reliable behavioral measure of stress and emotionality in rats [35,36,37]. Under basal conditions, male LVR rats displayed higher fecal counts after 20 min in a novel environment compared to male WT rats (4.2 ± 1.39 vs. 1.0 ± 0.52; *p* < 0.05) (Figure 2A).

Elevated adrenal wet weights and plasma corticosterone levels are other commonly used physiological markers of stress and increased HPA activity [45,46]. A two-way ANOVA for adrenal wet weights demonstrated a significant main effect of strain (F(2,27) = 23.4; *p* < 0.0001) and time (F(2,27) = 15.81; *p* < 0.0001). Sidak post-hoc analysis revealed that, compared to WT rats, adrenal wet weights were higher in LVR rats under both sedentary (394.67 ± 8.39 mg vs. 339.20 ± 8.39 mg; *p* < 0.05) and 4-week conditions (408.17 ± 11.57 mg vs. 355.20 ± 15.654 mg; *p* < 0.05; Figure 2B). Compared to sedentary levels, within- group adrenal wet weights were higher after 1 week of wheel-running for both WT (339.20 ± 9.65 mg vs. 433.70 ± 50.92 mg; *p* < 0.01) and LVR (390.75 ± 11.21 mg vs. 446.80 ± 7.95 mg; *p* < 0.01) rats (Figure 2B). After 4 weeks of wheel-running, adrenal wet weights were not significantly different from sedentary levels for either WT or LVR rats and were significantly lower compared to 1 week levels for both strains (WT: 413.70 ± 19.14 m vs. 355.20 ± 15.65 mg, *p* < 0.05; LVR: 457.00 ± 12.09 mg vs. 408.17 ± 11.57 mg; *p* < 0.05; Figure 2B).

A two-way ANOVA for plasma corticosterone levels revealed a significant main effect of strain (F(1,24) = 21.04; *p* < 0.0001) and time (F(2,24) = 5.26; *p* < 0.05). Plasma corticosterone levels were not different between LVR and WT under sedentary conditions (Figure 2C); however, LVR rats had significantly higher plasma corticosterone levels after 1 week of wheel access relative to WT (509.34 ± 49.27 ng/mL vs. 200.33 ± 62.60 ng/mL; *p* < 0.001; Figure 2C). Sidak post-hoc testing for within WT comparisons showed no significant difference in plasma corticosterone levels between sedentary and 1-week conditions. In contrast, corticosterone levels were significantly elevated after 1 week of running in LVR rats compared to either sedentary (509.34 ± 49.270 ng/mL vs. 274.46 ± 74.921 ng/mL; *p* < 0.01; Figure 2C), or 4-week levels (509.34 ± 49.27 ng/mL vs. 281.54 ± 45.32 ng/mL, *p* < 0.01; Figure 2C), with no difference between baseline and 4-week levels. This data supports that 1 week of wheel running increases markers of HPA activity, indicating a potential acute stress response to short-term PA that appears to normalize following chronic wheel access. Moreover, LVR rats appear to show a heightened stress response to acute running.

### 3.3. LVR Rats Show Increased Bla Expression of Pdyn following 1 Week of Wheel-Running Compared to Sedentary Levels

To test the effects of short-term versus long-term PA on aversion-related gene changes, BLA mRNA expression of *Pdyn*, *Oprk1*, and *Crf* were evaluated in WT and LVR rats that had remained sedentary or had voluntarily run for 1 or 4 weeks (Figure 3A–C). A two-way ANOVA for strain versus time for BLA *Pdyn* expression revealed a significant main effect of time (F(2,24) = 8.55; *p* < 0.01), but not strain. A Sidak post-hoc comparison showed no difference between all three WT groups; however, *Pdyn* was higher in LVR rats that had run for 1 week compared to sedentary values (1.99 ± 0.35 vs. 0.28 ± 0.06, *p* < 0.01), with no difference between sedentary and 4-week groups (Figure 3A). A two-way ANOVA for strain versus time for BLA *Oprk1* expression revealed a significant main effect of strain (F(1,24) = 6.37; *p* < 0.05), and a significant interaction (F(2,24) = 4.23; *p* < 0.05), but no effect of time. Interestingly, Sidak post-hoc comparisons showed significantly higher *Oprk1* expression levels in sedentary WT compared to sedentary LVR (1.13 ± 0.26 vs. 0.45 ± 0.083; *p* < 0.05), whereas 4 weeks of wheel running in WT trended to decrease *Oprk1* expression levels compared to sedentary conditions (1.131 ± 0.26 vs. 0.55 ± 0.09, *p* = 0.07; Figure 3B). Lastly, a two-way ANOVA for strain versus time for BLA *Crf* expression revealed a significant main effect of strain (F(1,24) = 7.84; *p* < 0.01) and significant strain by time interaction (F(2,24) = 9.69; *p* < 0.001), but no effect of time. Sidak post-hoc comparisons revealed that WT *Crf* expression was significantly higher at 1 week of wheel-running compared to LVR 1-week levels (1.19 ± 0.07 vs. 0.52 ± 0.07; *p* < 0.01). Moreover, WT BLA *Crf* expression was found to be significantly lower in the 4-week condition compared to the 1-week one (0.67 ± 0.12 vs. 1.20 ± 0.07, *p* < 0.05; Figure 3C).

### 3.4. LVR Rats Show Increased Levels of Phosphorylated Oprk1 Protein Levels following 1 but Not 4 Weeks of Wheel-Running

Lastly, to determine whether short-term versus long-term physical activity altered kappa opioid activity in the BLA (a molecular marker of aversion-related signaling), we evaluated phospho-OPRK1 protein levels from WT and LVR rats that were sedentary and that had voluntarily run for 1 or 4 weeks (Figure 4A,B). A one-way ANOVA for the effect of cage condition (Sed, 1 week, or 4 weeks of wheel-running) in WT rats revealed no main effects (Figure 4A). A one-way ANOVA for LVR cage condition revealed a trending main effect of cage condition (F(2,11) = 3.45; *p* = 0.06). However, Sidak post-hoc comparisons did show a significant, two-fold difference between sedentary and 1-week phospho-OPRK1 levels (1.00 ± 0.18 vs. 2.01 ± 0.35; *p* < 0.05; Figure 4B).

## 4. Discussion

Maintaining healthy and consistent levels of physical activity is a clinically proven and cost-effective means of reducing the onset of more than 40 chronic diseases [11,47]. However, daily stressors may act as a barrier to maintaining regular physical activity, especially in non-habitually active individuals [48]. To better evaluate the relationship between stress and physical activity, we sought to determine whether the stress of short-term physical activity (1) increases molecular markers of aversion in the BLA of male rats and (2) whether those markers were more pronounced in a unique, genetic model of low physical activity motivation. We show evidence for the latter, whereby our findings demonstrate the potential existence of causative genes for heightened stress reactivity following short-term PA in LVR rats. If identified, such mechanisms could help elucidate stress-related barriers to exercise adherence. More broadly, this work highlights that PA may be “aversive” in the short-term but may “normalize” when performed habitually in a genetic risk model for physical inactivity.

One week of wheel-running causes a hyperactivation of the corticosterone response in male Sprague-Dawley rats, with a return to baseline by 4 weeks of wheel-access [21]. Our current findings show that 1 week of wheel-running results in heightened adrenal wet weights and plasma corticosterone levels in LVR rats, which similarly appears to normalize after 4 weeks of wheel-running. Although there was no effect of wheel-running on plasma corticosterone levels in male WT rats, 1 week of wheel-running was associated with higher adrenal wet weights compared to either sedentary or 4 week conditions. These findings recapitulate evidence that acute voluntary PA acts as a physiological stressor, whereby it stimulates the HPA axis, increases glucocorticoid release, and alters stress- and aversion-related signaling in the brain [49]. Although many forms of stress are thought to cause glucocorticoid receptor downregulation in the pituitary, this effect is not seen in rats that voluntarily run for 4 weeks, despite showing an increase in plasma glucocorticoid levels [21,50]. By understanding the physiological and neurobiological responses to PA at these critical stages, and how these responses differ from other known stressors, we may begin to understand important neural mechanisms underlying the development and eventual reinforcement of PA and begin to identify important barriers to exercise adherence in humans.

LVR rats were selectively bred for the primary phenotype of voluntarily running low distances, whereby they reliably run ~20% of their founder line [26]. Despite their extraordinarily low PA levels, LVR rats find voluntary wheel-running reinforcing, yet appear less motivated to gain access to running wheels [24,26]. LVR rats also show lower sucrose preference compared to WT rats, further indicating potential reward and motivational deficits. Similar to adult rats that have experienced early-life social isolation, LVR rats display both reward and motivational deficits and high cAMP response element binding protein (CREB) activity in the nucleus accumbens [26]. Further, adult female LVR rats that experienced early-life social isolation ran significantly less than their group-housed and environmentally enriched littermates, whereas female Wistar rats showed no difference between socially isolated and enriched groups [26]. It is well established that socially isolated rats have heightened plasma corticosterone levels and show behavioral signs of anxiety, depression, and substance use; therefore, it follows that early-life stress may impact other patterned behaviors such as PA [26,51,52,53]. Our prior findings suggest that LVR rats may have a higher sensitivity to the effects of early-life social isolation and that their adult wheel-running behavior may be blunted as a result. In parallel, the present results show that LVR rats similarly have a higher stress sensitivity to acute wheel-running, whereby they display a more pronounced increase in adrenal wet weights and plasma corticosterone compared to WT rats. Taken together, these results suggest a generalizable effect of environmental stressors (social isolation and acute wheel-running) on HPA responsiveness in LVR rats and that their heightened sensitivity to known stressors may underlie a low motivation to be physically active. Importantly, future evaluations of the effects of stress on PA adherence requires careful consideration of both the genetic and environmental contributions.

The fact that so few individuals are habitually physically active, and when prescribed for health-related reasons, so few adhere to physical activity regiments, may have a basis in stress- and aversion-related neurobiology. For instance, systemic administration of CRF induces BLA kappa-opioid receptor phosphorylation and is thought to mediate anxiety-like behavior through the dynorphin/KOR system [54]. Here, WT rats showed elevated *Crf* expression relative to LVR following 1 week of wheel-running and appeared to normalize BLA *Crf* expression in response to chronic wheel-running. The lack of Crf response in LVR rats to 1 or 4 weeks of wheel running may result from a compensatory decrease in Crf given their higher baseline adrenal weights and emotionality responses to a novel environment. In support, other animal studies have shown similar compensatory decreases in brain *Crf* expression following neurectomy recovery and opiate withdrawal [55,56,57].

Prior characterization of distinct differences in dopaminergic, opioidergic, and glutamatergic signaling in the nucleus accumbens of LVR rats highlights the neural complexity and uniqueness of this genetic risk model for low PA motivation [42,43,58]. In the present study, WT rats that had run for 4 weeks had lower BLA *Oprk1* expression compared to sedentary levels, with no wheel-effect in LVR rats. Interestingly, *Oprk1* expression was lower in sedentary LVR compared to WT rats, which contrasts with earlier findings showing no difference in *Oprk1* expression in the nucleus accumbens [58]. Together, both sets of findings suggest a potential baseline difference in kappa-related mechanisms in LVR rats. Dynorphin/KOR-mediated signaling is thought to dampen neuronal activity in the accumbens and lead to behavioral sensitization. Moreover, dynorphinergic signaling in the shell region of the accumbens induces real time place preference, whereas stimulation of dynorphin-containing neurons in the accumbens core subregion causes robust aversive-like behaviors [59]. The observed differences in KOR expression may help to explain potential differences in the stress (namely adrenal wet weights) and emotionality responses of LVR and WT rats. However, it remains important to test these expression differences at the protein level and to further evaluate behavioral measures of anxiety and emotionality in these rats.

In all, the presented work reflects the known role of acute voluntary physical activity as a physiological stressor and highlights its potential importance in stimulating aversion-related signaling in the BLA. Our results may further reflect LVR rats as a genetic model of increased sensitivity to the stress- and aversion-related consequences of short-term physical activity. It was recently determined that LVR rats display anxiety-like behaviors and have higher expression of hippocampal gene markers associated with anxiodepressive behaviors [60]. Future research would benefit from addressing the role of dynorphin and kappa-related signaling events and signs of aversion-like behavior in response to short- versus long-term wheel-running, particularly in LVR rats. Because the dynorphin/KOR system plays such a critical role in the progression of alcohol use disorder and other substance abuse disorders, it follows that the addictive properties of voluntary physical activity would incorporate similar neuro-molecular mechanisms [61,62,63,64,65]. Addressing such questions would be another meaningful research direction with the goal of improving PA levels and exercise regimen adherence.

## Figures and Tables

**Figure 1 jfmk-08-00006-f001:**
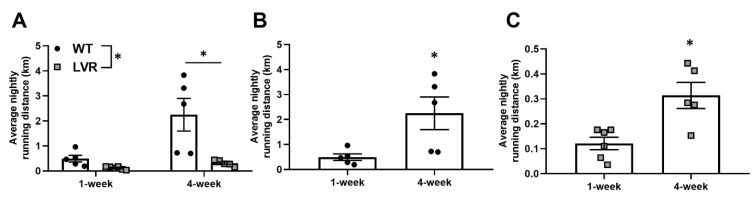
Phenotypic differences in wheel-running between wild-type (WT) (n = 5/running group; closed circles) and low voluntary runners (LVR) (n = 5–6/running group; gray squares) rats. (**A**) Average nightly running distances (km) for WT and LVR following 1 or 4weeks of voluntary wheel-access. (**B**) Average nightly running distance for WT rats that voluntarily ran for 1 or 4weeks. (**C**) Average nightly running distance for LVR rats that voluntarily ran for 1 or 4 weeks. * Denotes significant main effect (*p* < 0.05) of strain (**A**) or significant difference (*p* < 0.05) for Student’s *t* test (**B** and **C**). Values shown as mean ± SE.

**Figure 2 jfmk-08-00006-f002:**
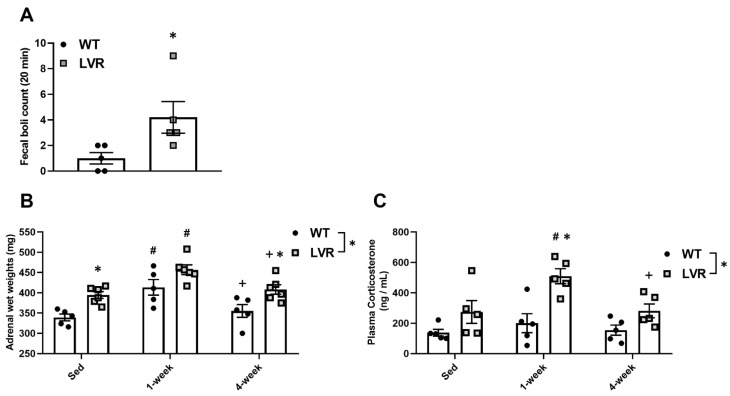
Differences in markers of stress-reactivity and HPA activity between WT (n = 5; closed circles) and LVR (n = 5–6; gray squares) rats. (**A**) Fecal boli counts following 20-min exposure to a novel environment under sedentary conditions. (**B**) Adrenal wet weights under sedentary, 1 week and 4 week wheel-running conditions (mg). (**C**) Plasma corticosterone levels under sedentary, 1week and 4 week wheel-running conditions (ng/mL). * Denotes main effect of strain and post-hoc differences (*p* < 0.05) between groups within timepoint. # post-hoc difference (*p* < 0.05) relative to within group sedentary animals. + post-hoc difference 4 weeks vs. 1 week within group (*p* < 0.05). Values shown as mean ± SE.

**Figure 3 jfmk-08-00006-f003:**
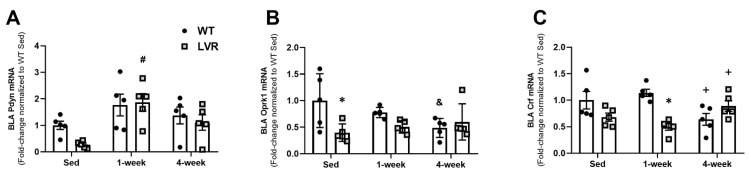
Effect of 1 versus 4 week(s) of wheel-running on BLA mRNA expression of (**A**) *Pdyn*, (**B**) *Oprk1,* and (**C**) *Crf* in WT (n = 5/group) and LVR (n = 5/group) rats compared to sedentary controls. ***** Denotes significant post-hoc difference (*p* < 0.05) between groups within timepoint. # post-hoc difference (*p* < 0.05) relative to within group sedentary animals. + post-hoc difference 4 weeks vs. 1 week within group (*p* < 0.05) and post-hoc difference 4 weeks vs. sed (*p* < 0.05). Values shown as mean ± SE.

**Figure 4 jfmk-08-00006-f004:**
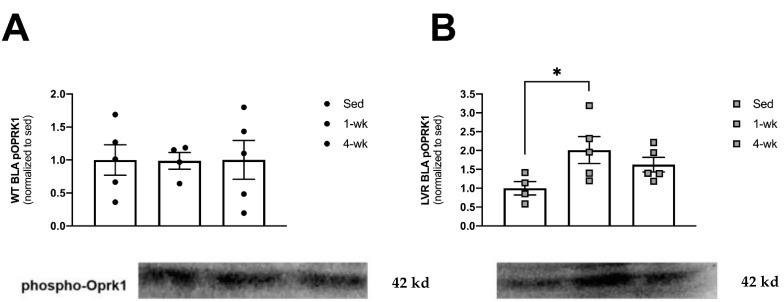
Comparison of sedentary versus 1 or 4 weeks of wheel-running on BLA phospo-Oprk1 protein expression in (**A**) WT (n = 5/group; closed circles) and (**B**) LVR (n = 5/group; gray squares). * post-hoc difference relative to sedentary group (*p* < 0.05). Values shown as mean ± SE.

**Table 1 jfmk-08-00006-t001:** Primer sequence for basolateral amygdala gene expression analyzed by qPCR.

Gene	Forward (5′–3′)	Reverse (3′–5′)	Accession Number
18s	GCCGCTAGAGGTGAAATTCTTG	CATTCTTGGCAAATGCTTTCG	NR_046237
Pdyn	AACTGCCATAGGGGGATTTGG	GGATGGCCGATCCAAGATTCA	NM_019374
Oprk1	AAACATCAGGGACGTGGACC	CTCCCTTCCCAAATCAGCGT	XM_032906449
Crhr1	AAGGCTACCAGACTTGCTCG	GGGCTTCGCACCCTTCC	NR_126013

## Data Availability

Not applicable.

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
