# Peer review of "Acute Wheel-Running Increases Markers of Stress and Aversion-Related Signaling in the Basolateral Amygdala of Male Rats"

_jfmk, 2022, doi:10.3390/jfmk8010006_

Round 1

Reviewer 1 Report

This study implemented a rat model of physical activity (wheel running) to evaluate whether stress is a possible cause for low physical activity. Some biomarkers for stress were measured in the periphery as well as within a brain region that is well-known for promoting stress, anxiety, and aversion – the basolateral nucleus of the amygdala (BLA). In brief, acute physical activity was associated with stress-related markers in the periphery, such as elevated adrenal weights and corticosterone levels, especially in rats bred for low voluntary running (LVR). Physical activity also seemed to elevate expression of prodynorphin (Pdyn) and phosphorylation of kappa opioid receptors (p-OPRK1) in the BLA, again in a more prominently manner in LVR rats compared to wild-types. Other changes and/or group differences were also detected in response to physical activity, for example in BLA CRF levels. Altogether, the authors concluded that such alterations on stress-related signals by physical activity could explain low motivation for physical activity, and that could also shed some light onto typically observed barriers for exercise adherence in humans.

Overall, this study is straightforward. It offers a thought-provoking perspective on how stress-related signals may diminish physical activity. This topic is of great interest for the general public. The text is well written, and the figures are easy to understand. Yet, I have some suggestions for improvements (see below).

Issues for Revisions:

#1: Regarding statistical analyses, there are inconsistencies in post-hoc tests. Sometimes Sidak was used, sometimes Tukey, sometimes Dunnett. There is no clear explanation or justification for using one post-hoc test versus the other. On some occasions, the type of post-hoc test was not specified (e.g., P.6, #252: “Interestingly, post-hoc comparisons showed significantly higher Oprk1 252 expression levels…”).

#2: More on statistical analyses, the methods mentioned that “All values are presented as mean ± standard error (SE)”. However, there is no mention anywhere whether normality tests were performed to determine whether the data followed normal distributions. This could be easily done in GraphPad Prism.

#3: In Fig.3, all data was normalized to the WT-Sed group. If so, why in Fig.3B, the value for the WT-Sed group is higher than 1.0? It should be exactly 1.0, right? Maybe there’s some error in the calculation.

#4: Corticosterone is greatly influenced by the circadian rhythm. However, there is no mention whether the authors used a consistent time window across groups for collecting blood samples for Cort measurements.

Author Response

Thank you for taking the time to review our manuscript and we appreciate your very helpful feedback.

Response to reviewer comments:

  1. These inconsistencies have now been corrected for. Sidak post-hoc analysis has been used for all ANOVA calculations and this did not alter the significance for any of the data shown.
  2. Thank you for pointing this out. All ANOVA tests assume a Guassian distribution and have undergone and passed the D'Agostino-Pearson normality test. This has now been specified in lines 138-139.
  3. This is an issue that can occur when using the 2^delta delta CT method when the control group has high variability. The math does not always work out perfectly through one when calculating fold change. However, I have now taken the groups and renormalized them all relative to one which adjusts for this shift in fold change and properly sets the control group to 1 exactly. Figure 3 has now been replaced reflecting this change.
  4. Thank you for bringing this to our attention, we have now added a line clarifying that blood was taken for all experimental animals within a one hour time window (line 101).

Reviewer 2 Report

Authors described the effect of acute physical training (for 1 week) and  chronic training (4 weeks) on the stress markers in Wistar male rats and low voluntary wheel- running (LVR) rats. Stress markers were assessed as plasma corticosterone level, adrenal wet gland and emotional behavior in experiment 1, and as KOR, dynorthin and CRF mRNA expression and OPRK1 protein phosphorylation in basolateral amigdala (BLA).  

Unfortunately, authors did not provide comprehensive discussion of the obtained results especially in cases when they contradict to other studies. Discussion of the results does not provide sufficient rational for conclusion statement about the role of acute physical activity as physiological stressor and its importance in stimulation aversion-related signaling in BLA, while the LVR rats indeed demonstrate stress reaction to acute physical activity and could be used as a model for the study of underlying biological mechanisms. 

Bellow listed specific comments related to the manuscript.  

Introduction 

·      Line 45: “Habitual PA, like other behavioral coping strategies (overeating, substance and alco-45 hol use, etc.), is considered physiologically stressful.” Probably, authors mean here the acute PA rather than habitual, regular PA that has been shown to increase the stress resilience. 

·      Line 80-87: Authors describes two sets of experiments, that later in discussion are described as experimental goals. Please, clarify, whether you have designed two separate studies, or data were collected from the same study to test the functional/peripheral stress reaction, and CNS markers?

Material and Method:

·      Line 110-116 describe experiment 1 and experiment 2, while line 108-110 “After which, blood, adrenal 108 glands, and brains were rapidly removed and 2-mm diameter BLA punches were imme-109 diately frozen in liquid nitrogen and kept at -80 °C until processing” gives impression that samples were collected from the same animals.

·      Experimental schedule and timeline is not clear. When were samples collected regarding last physical activity? Was all data presented in the study were collected from the same or different set of the animal. If different (2 separate experiments), then what is the reason to use extra animals when all samples could be collected with the use of smaller number of animals? 

·      Why was emotionality assessment done only for naïve (sedentary) animals? When authors discuss later about stress sensitivity of LVR line, it would be very useful to show the functional data from behavioral assessment. Why authors did not use opportunity to study another parameters of emotional behavioral when release animals to novel environment for fecal boli count? The design of the test is similar to open field test popular test for anxiety-behavioral assessment and could provide additional useful information about animal reaction to stressful situation.  How animals followed the 3R principals?

·      Corticosterone level depends ion circadian activity, at what period of time the samples were collected?

·       What is the rational for statistical method selection? Why similar set of data was tested by two way ANOVA with Sidak post-hoc for analysis of  running distance, corticosterone level, gland weight,  two way ANOVA with Tukey post-hoc for and mRNA expression,  and one way ANOVA separately for two strains for protein level? Authors should use the same principal for method selection for all data.

·      Data were not analysed for normality distribution. Statistical methods described in the manuscript could be used for normally distributed data, while some data obviously have non-normal distribution.

Data

·      Figure 1 demonstrate the same data in 3 different panels using two different statistical method. Statistical method should be justified.

·      Line 236-237 “  This data supports that 1-week of wheel running increases markers of HPA activity, indicating a potential acute stress response to short-term PA that appears to normalize following chronic wheel access” contradicts the results described just few lines above (231-232) demonstrating  absence of significant difference in WT group between sedentary and week 1.

Conclusions

·      Line 292-294. Authors claims that manuscript show evidence for  goal 1) described as “to determine whether the stress of short-291 term physical activity 1) increases molecular markers of aversion in the BLA of male rats”. It contradicts  to the data demonstrated absence of BLA markers (both mRNA expression and  OPRK1 protein phosphorylation) in WT rats (Figure 3 and 4). All further discussion is based on that contradiction statement.

·      Big part of the discussion takes the description of the authors previous study on effect of social isolation in LVR rats. The connection between effect of social isolation and described effect of acute PA in LVR rats is absolutely unclear. 

References

·      Three reference 44,45, 47 relate to authors previous publication describing the same method for BLA mRNA isolation and protein extraction. What is the rational of referring to three different publications?

·      Paper Grigsby 2018 was listed three times (24, 26 and 43), Grigsby 2020 – two times (59 and 27). 

·      In the discussion (line 320-328), wrong style of reference was used. 

·      References list and reference style should be checked by authors before submission. 

Author Response

Thank you for taking the time to review our manuscript and provide your very helpful feedback; it is greatly appreciated!

Response to reviewer:

1) Comment on Line 45: Corrected to clarify acute PA is physiologically stressful.

2) Comment on Line 80-87: Thank you for bringing this discrepancy to our attention. We have now clarified that all data was collected from the same study (lines 95-101)

3) Comment on Line 110-116: We have clarified that all experiments were done on the same set of animals (line 100).

4) We have attempted to clarify the experimental timeline by adding that animals retain wheel-access up until time of sacrifice which occurred within one-hour of lights-on in the animal housing room (line 103).

5) Regarding your question on why novel environment testing was only done on sedentary animals, we have added in the discussion that future work should further evaluate behavioral measures of anxiety in these rats (line 294). We did not do novel environment exposure in the wheel access groups as it added a confounding variable at time of sacrifice. If we did novel environment testing at 1 and 4 weeks of wheel running and then sacrificed the rats at those same timepoints in order to keep the wheel access times accurate it would likely result in an exaggerated molecular signature of anxiety in our wheel access rats. So we chose to not expose them to any external stressful environment that may alter their anxiety levels other than wheel access.

6) We have now clarified that we collected corticosterone levels within one hour of lights on which is a known trough in diurnal plasma corticosterone levels (line 104).

7) The listing of Tukey rather than Sidak was accidental and a mistake on our part. Sidak post hoc analysis has been used for all ANOVA's. For the pOPRK1 protein data, they were ran as a 1-way ANOVA because the LVR and WT samples had to be ran on separate blots due to the total number of samples. Therefore, we could only make within group analysis which is why we performed a 1-way ANOVA instead of a 2-way.

8) Regarding your comment on normal distribution, we have now added a line clarifying that all data undergoing ANOVA analysis were also confirmed to have a Gaussian distribution and passed a D'Agostino-Pearson normality test (lines 140-141).

9) Figure 1 was separated into 3 different panels to emphasize both the running difference between LVR and WT and that both WT and LVR increase their daily running over time. Panels B and C were split up due to the major difference in nightly running distance. Since LVR run a fraction of what WT do, it looked best graphically for them to be graphed separately. A student's t test was used for B and C as figure A already demonstrated that there is a significant difference between LVR and WT running distances.

10) While WT rats did not have a significant increase in plasma corticosterone levels after 1 week of wheel access, they did have a significant increase in adrenal wet weights which would support that HPA activity is increased in WT as well, albeit not to the same extent as LVR. We tried to clarify this further in the discussion by emphasizing that most of the effects were seen in LVR rather than WT (lines 239-240).

11) We agree with your point and have clarified by adding a line in the discussion stating that our data provides potential causative genes for heightened stress reactivity following short term physical activity in LVR rats, thereby specifying that these effects are primarily seen in LVR rats rather than WT rats (lines 239-240).

12) We have expanded on the connection between our current data and previous data on social isolation that should clarify the connection between the two (lines 263-273).

13) Thank you for bringing this to our attention, we are not sure how these citation repeat issues happened and apologize for the mistake. All citation and reference issues have now been fixed. All duplicated references have been removed and in text citations are updated to properly match the correct references.